# Cancer type classification using plasma cell-free RNAs derived from human and microbes

Shanwen Chen[1,2†], Yunfan Jin[3†], Siqi Wang[3†], Shaozhen Xing[3†], Yingchao Wu[1], Yuhuan Tao[3], Yongchen Ma[1], Shuai Zuo[1], Xiaofan Liu[3], Yichen Hu[4], Hongyan Chen[5], Yuandeng Luo[6], Feng Xia[6], Chuanming Xie[6], Jianhua Yin[7], Xin Wang[8], Zhihua Liu[5], Ning Zhang[2], Zhenjiang Zech Xu[4,9,10]*, Zhi John Lu[3]*, Pengyuan Wang[1]*

[1]Division of General Surgery, Peking University First Hospital, Beijing, China; [2]Translational Cancer Research Center, Peking University First Hospital, Beijing, China; [3]MOE Key Laboratory of Bioinformatics, Center for Synthetic and Systems Biology, School of Life Sciences, Tsinghua University, Beijing, China; [4]State Key Laboratory of Food Science and Technology, Nanchang University, Nanchang, China; [5]State Key Laboratory of Molecular Oncology, National Cancer Center/National Clinical Research Center for Cancer/Cancer Hospital, Chinese Academy of Medical Sciences and Peking Union Medical College, Beijing, China; [6]Institute of Hepatobiliary Surgery, The First Hospital Affiliated to Army Medical University, Chongqing, China; [7]Department of Epidemiology, Faculty of Navy Medicine, Navy Medical University, Shanghai, China; [8]Department of Breast Surgical Oncology, National Cancer Center/ National Clinical Research Center for Cancer /Cancer Hospital, Chinese Academy of Medical Sciences and Peking Union Medical College, Beijing, China; [9]Shenzhen Stomatology Hospital (Pingshan), Southern Medical University, Shenzhen, China; [10]Microbiome Medicine Center, Department of Laboratory Medicine, Zhujiang Hospital, Southern Medical University, Guangzhou, China

*For correspondence:
zhenjiang.xu@gmail.com (ZZX);
zhilu@tsinghua.edu.cn (ZJohnL);
pengyuan_wang@bjmu.edu.cn
(PW)

†These authors contributed
equally to this work

Competing interest: The authors
declare that no competing
interests exist.

Reviewing Editor: YM Dennis
Lo, The Chinese University of
Hong Kong, Hong Kong

**Abstract** The utility of cell-free nucleic acids in monitoring cancer has been recognized by both scientists and clinicians. In addition to human transcripts, a fraction of cell-free nucleic acids in human plasma were proven to be derived from microbes and reported to have relevance to cancer. To obtain a better understanding of plasma cell-free RNAs (cfRNAs) in cancer patients, we profiled cfRNAs in ~300 plasma samples of 5 cancer types (colorectal cancer, stomach cancer, liver cancer, lung cancer, and esophageal cancer) and healthy donors (HDs) with RNA-seq. Microbe-derived cfRNAs were consistently detected by different computational methods when potential contaminations were carefully filtered. Clinically relevant signals were identified from human and microbial reads, and enriched Kyoto Encyclopedia of Genes and Genomes pathways of downregulated human genes and higher prevalence torque teno viruses both suggest that a fraction of cancer patients were immunosuppressed. Our data support the diagnostic value of human and microbe-derived plasma cfRNAs for cancer detection, as an area under the ROC curve of approximately 0.9 for distinguishing cancer patients from HDs was achieved. Moreover, human and microbial cfRNAs both have cancer type specificity, and combining two types of features could distinguish tumors of five different primary locations with an average recall of 60.4%. Compared to using human features alone, adding microbial features improved the average recall by approximately 8%. In summary, this work provides evidence for the clinical relevance of human and microbe-derived plasma cfRNAs and their potential utilities in cancer detection as well as the determination of tumor sites.

## Editor's evaluation

This study provides an interesting clinical relevance of human and microbe cell free RNAs derived from plasma that can be used as biomarkers for cancer detection and cancer type classification, and thereby having potential in clinical application.

## Introduction

Recently, noninvasive liquid biopsy of plasma cell-free nucleic acids has emerged as a convenient and cost-effective method for cancer screening and monitoring. The clinical utilities of cell-free DNA (cfDNA) and cell-free RNA (cfRNA) in cancer have been extensively studied. Mutations (*Abbosh et al., 2017*), methylation levels (*Anders et al., 2015*), fragmentation patterns (*Cristiano et al., 2019*) of plasma cfDNA, and expression levels of different cfRNA species (miRNA, circular RNA [circRNA], signal recognition particle RNA [srpRNA], long noncoding RNA [lncRNA], mRNA, etc.) (*Best et al., 2015*; *Li et al., 2015*; *Tan et al., 2019*) in plasma, platelets, and extracellular vesicles (EVs) were identified as potential diagnostic or prognostic markers. In addition to early detection, it is also favorable if liquid biopsy could provide clues about the tumor's primary location to guide further clinical decisions. Plasma cfDNA methylation and the platelet transcriptome were reported to have cancer type specificity (*Shen et al., 2018*; *Best et al., 2015*) but whether plasma cfRNAs have such properties remains largely uncharacterized.

Studies of the human cancer-related microbiome are increasingly valued for their novel biological insights and potential clinical applications. It is well established that several bacteria and viruses are involved in cancer development and progression. For instance, chronic infection with HBV and HPV is the leading cause of liver cancer and cervical cancer, respectively (*Arbuthnot and Kew, 2001*; *Burd, 2003*). *Helicobacter pylori* infection is a well-known risk factor for developing gastric cancer (*Polk and Peek, 2010*). *Fusobacterium nucleatum* was reported to drive tumorigenesis in colon cancer (*Han, 2015*). It has also been reported that in pancreatic cancer, higher microbial diversity predicts better prognosis (*Riquelme et al., 2019*). A more recent study reported that cancer type-specific living bacteria can be detected inside tumor cells, suggesting that there are unexpectedly complicated interactions between microbes and tumor cells (*Riquelme et al., 2019*).

Traditionally, blood was thought to be sterile in individuals without sepsis (*Gosiewski et al., 2017*; *Blauwkamp et al., 2019*). Although it remains controversial whether the blood of healthy donors (HDs) contains living bacteria (*Best et al., 2015*; *Potgieter et al., 2015*), several recent studies suggested that bacteria-derived nucleic acids can be confidently detected in human plasma, which cannot be simply attributed to contamination in reagents and other potential sources (*Gosiewski et al., 2017*; *Zozaya-Valdés et al., 2021*; *Kowarsky et al., 2017*; *Pan et al., 2017*). Many uncharacterized bacteria and viruses can be assembled from blood DNA-seq data (*Kowarsky et al., 2017*). In obese patients, gut microbe-derived EVs, which contain microbial DNA, can enter the bloodstream and induce an inflammatory response (*Luo et al., 2021*). A recent study also suggested that the abundance of microbial-derived plasma cfDNA could accurately distinguish between different cancer types (*Poore et al., 2020*).

Most of the previous cfRNA studies focused on small RNA species (*Mitchell et al., 2008*), which are relatively stable in plasma. Long RNA species in plasma have relatively low concentrations, which are mainly 100–200 nt fragments lacking poly-A tails and intact ends. Therefore, regular RNA-seq, which usually uses ligation techniques to add adaptors, will not work well for long cfRNAs. The recently developed SMART-seq (*Picelli et al., 2014*)-based techniques offer the potential to overcome these issues. Furthermore, to sequence total RNAs in plasma, we need to simultaneously remove the abundant rRNA fragments, which are enabled by a CRISPR-based technology called depletion of abundant sequences by hybridization (DASH; *Gu et al., 2016*). This motivated us to study the biological relevance and clinical utilities of human and microbe-derived long cfRNAs, taking advantage of the above techniques.

Here, we investigated diverse cfRNA species (>50 nt, rRNA depleted) in ~300 plasma samples of cancer patients and HDs. This cohort included five cancer types (colorectal cancer, stomach cancer, liver cancer, lung cancer, and esophageal cancer) that were responsible for 75% of cancer-related mortality in China (*Siegel et al., 2015*). Most of the cancer patients were in the early stages. To the best of our knowledge, our study demonstrated for the first time that both human and microbe-derived RNAs in

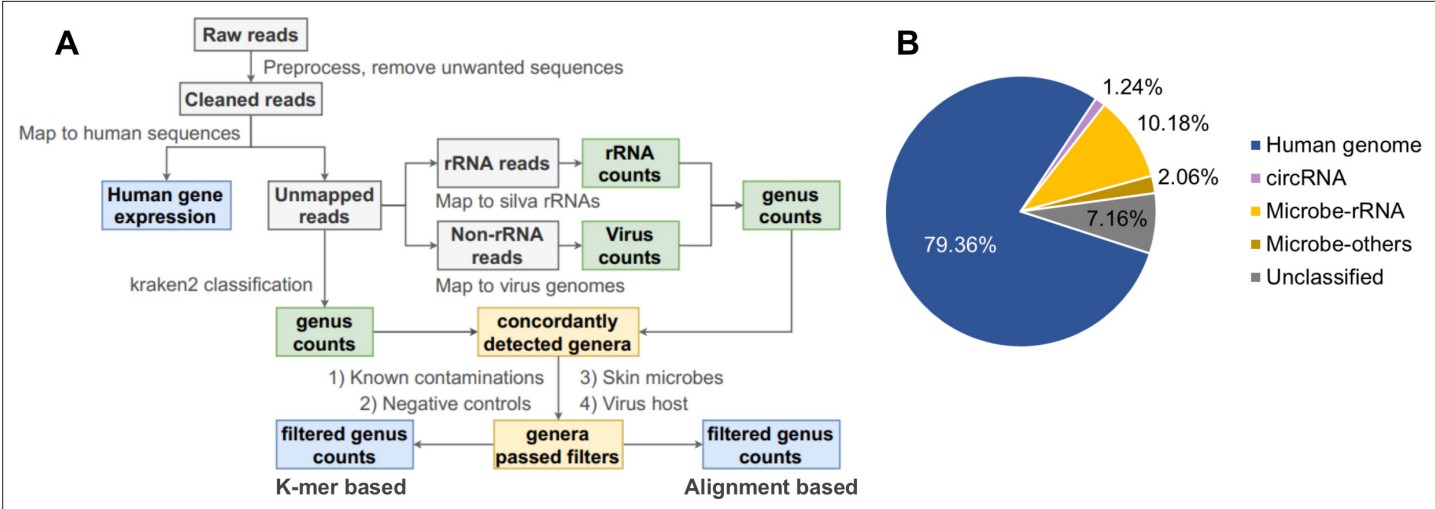

**Figure 1.** Pipeline for cell-free RNA (cfRNA) sequencing data processing. (**A**) The bioinformatic pipeline for plasma cfRNA sequencing data processing. After adapter trimming, spike in, potential vector contaminations, and human rRNA sequences were removed. Cleaned reads were aligned to the human genome and circular RNA back-spliced junctions. Unmapped reads were classified with a k-mer-based pipeline and an alignment-based pipeline. Genera detected by both pipelines were used for downstream analysis. Potential contaminations (known common laboratory contaminants, genera detected in control samples, skin microbes, and suspicious viral genera) were excluded. See the Materials and methods section for details. (**B**) Average fractions of different cfRNA components in cleaned reads. Microbe-rRNA refers to reads annotated to rRNA. Microbe-others refers to non-rRNA reads that were assigned to microbial genomes by kraken2.

The online version of this article includes the following figure supplement(s) for figure 1:

**Figure supplement 1.** Quality control of sequencing data.

plasma detected by cfRNA-seq could reflect cancer type-specific information. We also showed that combining microbial cfRNA signatures could improve the performance of human cfRNAs in cancer classification.

## Results

### Sequencing of cfRNAs captures signals of various long RNA species in the plasma

Here, we adapted a SMART-based total RNA sequencing method (SMART-total) to profile plasma total cfRNAs. This technique was optimized for low-input RNA sequencing and robust for partially degraded RNA fragments. SMART-total was successfully applied to detect cfRNAs in the plasma of pregnant women and cancer patients in previous studies (*Pan et al., 2017*; *Ngo et al., 2018*; *Yu et al., 2020*). One of these studies, which investigated plasma cfRNAs of pregnant women, suggested that microbial signals detected by SMART-total can also provide useful information (*Pan et al., 2017*). We applied SMART-total to a cohort of 295 plasma samples, and the percentage of patients with early-stage cancer (stages I and II) ranged from 65% in stomach cancer to 86% in lung cancer (*Supplementary file 1*).

For low-biomass metagenomic profiling, laboratory and kit contamination can lead to unreliable conclusions (*Eisenhofer et al., 2019*). Given the low concentration of both human and microbial cfRNAs in plasma, little contamination could have detrimental impacts on downstream analysis. To minimize the impacts of potential microbe contamination introduced in sample collection, RNA extraction, library preparation, and sequencing, two *Escherichia coli* samples and one human brain RNA sample were processed and sequenced following exactly the same procedure as plasma samples, serving as controls for contamination.

In addition to potential contaminations, misclassification of microbe-derived reads also renders the result less interpretable. We carefully designed a computational pipeline to mitigate these problems (*Figure 1A*, see Materials and methods). In brief, after removing human rRNA and other unwanted sequences, reads were aligned to the human genome and circRNA back-spliced junctions to quantify

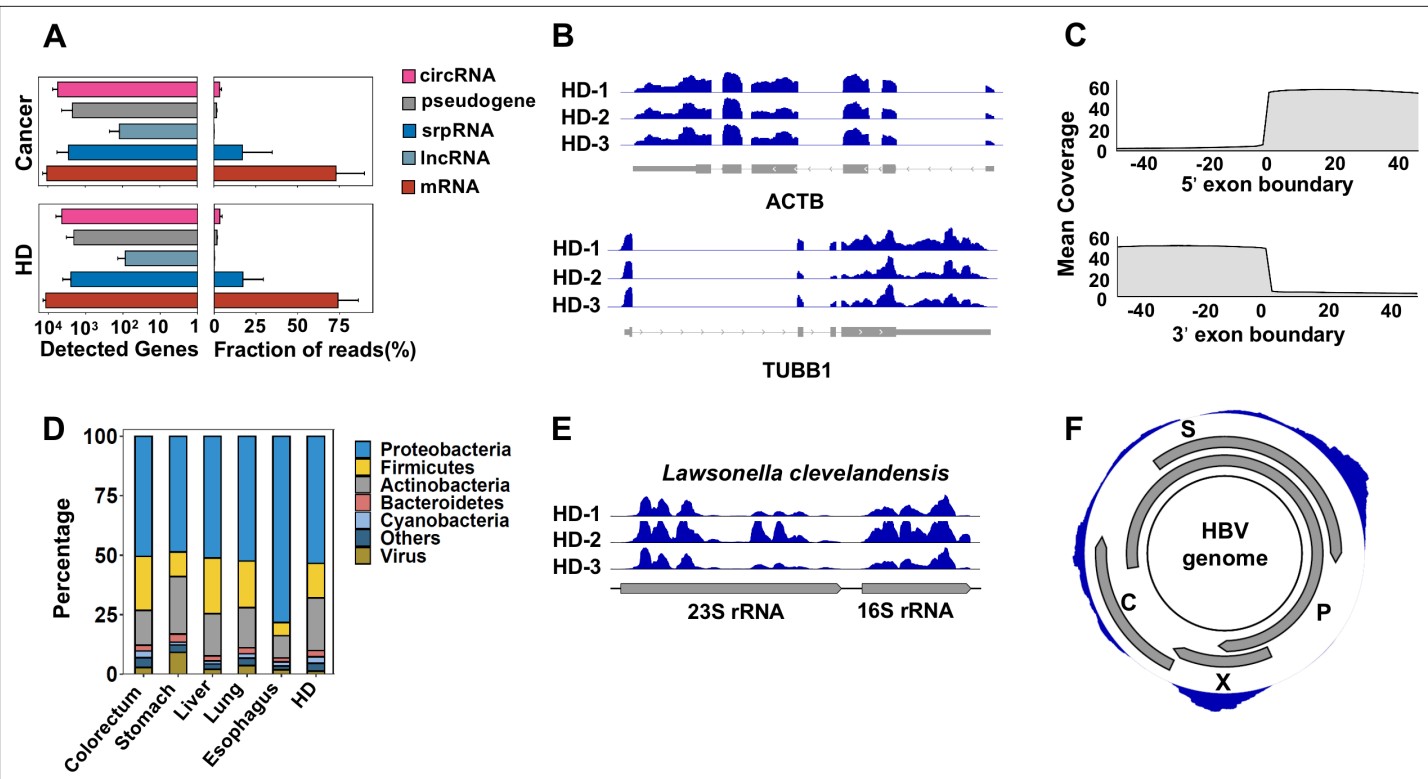

**Figure 2.** Human genes and microbial signals revealed by cell-free RNA (cfRNA)-seq. (**A**) The number of detected human transcripts (counts per million >2) of different RNA types and their relative abundances. (**B**). Representative coverages for ACTB and TUBB1 in healthy donors (HDs) from three clinical centers (samples HD-1, HD-2, and HD-3 are provided by PKU, ShH-1, and SWU, respectively). (**C**). Metagene plot for read coverage around 5' exon boundaries and 3' exon boundaries. The mean coverage of 100 nt around exon boundaries for exons with read coverage >3 is shown. (**D**). Relative abundance of reads assigned to different phyla by kraken2. (**E**). Representative read coverage of *Lawsonella clevelandensis* 16S and 23S rRNA in healthy donors from three clinical centers. (**F**). A representative read coverage on the HBV genome in cfRNA of a patient with liver cancer.

The online version of this article includes the following figure supplement(s) for figure 2:

**Figure supplement 1.** Most abundant human genes and microbial genera in plasma cell-free (cfRNA) libraries.

human gene expression. Several quality control rules were applied to ensure data reliability, and 263 high-quality samples were reserved for further analysis (*Figure 1—figure supplement 1*, *Supplementary file 2*). Unaligned reads were classified with kraken (*Wood et al., 2019*), an efficient but less stringent method based on k-mer contents and a stringent but relatively computationally intensive method based on bowtie2 alignment (*Langmead and Salzberg, 2012*). Since the majority of microbial reads are rRNA, we only mapped microbial rRNA reads against the Silva database (*Yilmaz et al., 2014*) to reduce the computational burden. The rest non-rRNA reads were aligned to viral genomes. From the resulting microbial profile, we filtered away genera that were found in our control samples (*Supplementary file 3*), previously reported common laboratory contaminations (*Salter et al., 2014*), and abundant skin microbes (*Oh et al., 2016*), which are often regarded as potential sources of contamination (*Schierwagen et al., 2020*). Several suspicious viral genera with nonhuman eukaryotic hosts (*Mihara et al., 2016*) were also excluded (*Supplementary file 3*).

Using this computational pipeline, the majority of cleaned reads were mapped to the human genome (79.36% on average) and back-spliced junctions of circRNA (1.24% on average). In the remaining reads, 10.18% were annotated as nonhuman rRNA, and 2.06% were further assigned to microbial genomes by kraken2 (*Figure 1B*, *Supplementary file 4*).

Consistent with the intracellular long RNA profile, mRNAs and lncRNAs were the most abundant human RNA species captured in the SMART-total library (*Figure 2A*). Several housekeeping genes, such as ACTB, TUBB1, and PTMA, as well as noncoding RNAs, such as srpRNA (RN7SL2), are highly abundant in the plasma of both cancer patients and HDs (*Figure 2—figure supplement 1*). For these transcripts, the coverage was uniformly distributed along the full-length transcripts in samples from

different clinical centers (*Figure 2B*). Previous studies demonstrated that mRNAs mainly exist as short fragments up to several hundred nucleotides (*Larson et al., 2021*). This uniform coverage indicates that at least for these most abundant transcripts, such a naturally occurring fragmentation process does not have a strong sequence preference. Moreover, a sharp boundary of read coverage at exon-intron junctions further demonstrated that there was minimal genomic DNA contamination in our sequencing libraries (*Figure 2C*).

For microbe-derived reads, the most abundant phylum was *Proteobacteria*, followed by *Firmicutes* and *Actinobacteria* (*Figure 2D*). This composition resembles previous reports for microbe-derived cfDNA and cfRNA in plasma (*Zozaya-Valdés et al., 2021*; *Pan et al., 2017*; *Yao et al., 2020*; *Paisse et al., 2016*; *Lelouvier et al., 2016*). Consistent with previous studies (*Liang and Bushman, 2021*), *Caudovirales*, an order of viruses known as tailed bacteriophages, makes up the majority (the median fraction is higher than 95%) of reads assigned to viruses by kraken2.

We investigated the read coverage for detected microbes by aligning nonhuman reads to their genomes. As expected, for bacteria, most of the RNA-seq signals agree with the previous notion that most microbial reads are from rRNA, and for *Lawsonella clevelandensis*, a pathogen reported to induce abscess (*Goldenberger et al., 2019*) as an example (*Figure 2E*). The RNA-seq signals for viruses are also consistent with their genome annotations. For instance, in a representative coverage of the HBV genome (*Figure 2F*), the read coverage of HBX gene agrees well with its annotated boundary.

**Table 1.** Enriched Kyoto Encyclopedia of Genes and Genomes (KEGG) pathways of significantly up- and downregulated human genes based on the cell-free RNA (cfRNA)-seq data of all cancer patients vs. healthy donors (HDs).

| Pathways[*] | p value | Gene ratio[†] | Trend |
|---|---|---|---|
| Platelet activation | 1.52E-19 | 0.0728 | Up |
| Calcium signaling pathway | 1.28E-07 | 0.0695 | |
| ECM-receptor interaction | 2.98E-07 | 0.0364 | |
| Neutrophil extracellular trap formation | 4.15E-07 | 0.0579 | |
| Focal adhesion | 5.79E-07 | 0.0596 | |
| Phospholipase D signaling pathway | 3.52E-06 | 0.0464 | |
| Human cytomegalovirus infection | 8.81E-06 | 0.0596 | |
| Regulation of actin cytoskeleton | 1.09E-05 | 0.0579 | |
| Rap1 signaling pathway | 1.21E-05 | 0.0563 | |
| Viral carcinogenesis | 4.16E-05 | 0.0530 | |
| Ribosome | 2.32E-69 | 0.174 | Down |
| PD-1 checkpoint pathway in cancer | 1.74E-03 | 0.026 | |
| Proteasome | 2.51E-03 | 0.017 | |
| Pyrimidine metabolism | 3.43E-03 | 0.019 | |
| Th17 cell differentiation | 3.85E-03 | 0.028 | |
| Th1 and Th2 cell differentiation | 6.45E-03 | 0.025 | |
| Transcriptional misregulation in cancer | 6.85E-03 | 0.042 | |
| NOD-like receptor signaling pathway | 7.08E-03 | 0.040 | |
| Cytosolic DNA-sensing pathway | 7.16E-03 | 0.019 | |
| NF-kappa B signaling pathway | 7.39E-03 | 0.026 | |

[*]Enriched pathway of pan-cancer upregulated and downregulated genes.
[†]Fraction of upregulated or downregulated genes annotated to a KEGG pathway.

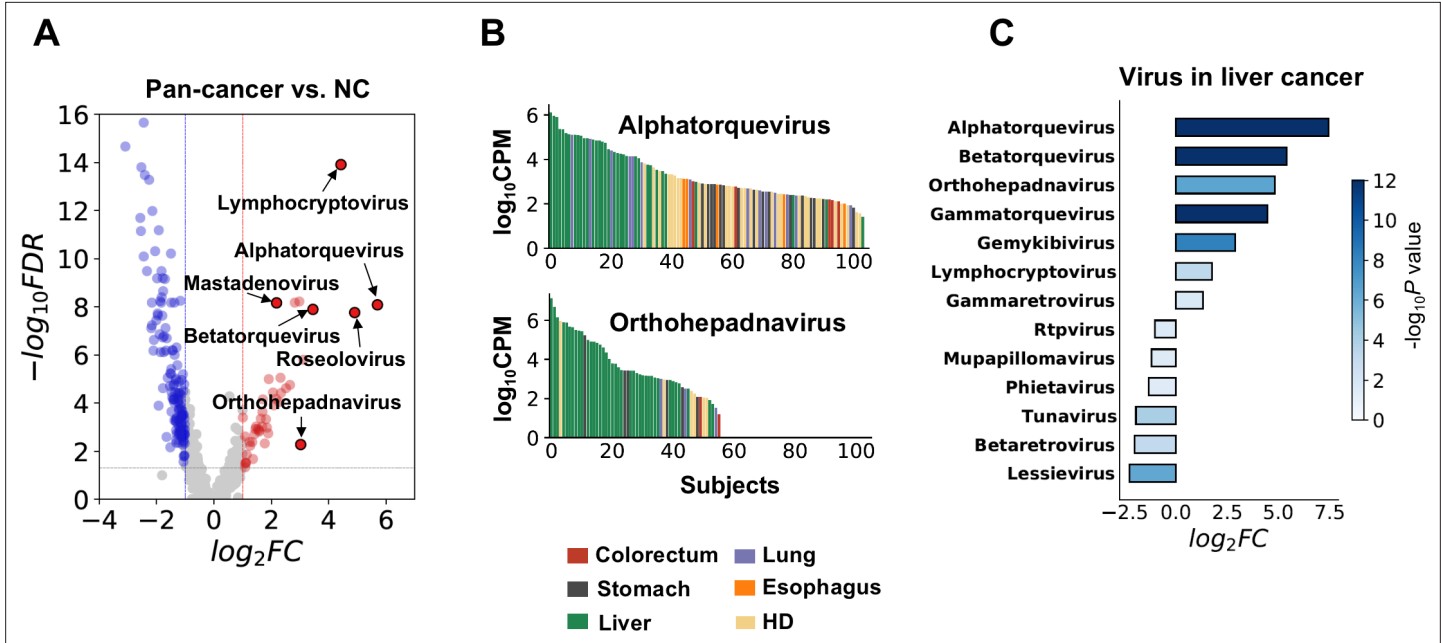

**Figure 3.** Biological relevance of alterations in the microbial cell-free RNA (cfRNA) profile. (**A**) Example genera with significantly altered abundance in cancer patients when compared to healthy donors (HDs). FC: fold change. FDR: false discovery rate. FC and FDR were calculated using the result of the alignment-based method, and labeled genera were supported by both pipelines. (**B**) Abundance of *Alphatorquevirus* and *Othohepavirus* in the alignment-based pipeline across different samples ranked in descending order; colors indicate different sample groups. (**C**) Virus genera with significant abundance alterations (FDR <0.05 and log₂fold-change >1) in liver cancer patients when compared to HDs.

The online version of this article includes the following figure supplement(s) for figure 3:

**Figure supplement 1.** Enriched Kyoto Encyclopedia of Genes and Genomes (KEGG) pathways of differentially expressed human genes for each cancer type.

## cfRNA profile alterations in patients are cancer relevant

To investigate the biological relevance of plasma cfRNAs in cancer patients, the enriched Kyoto Encyclopedia of Genes and Genomes (KEGG) pathways of human genes differentially expressed in cancer patients (*Supplementary file 5*) were identified (*Table 1*). Enriched pathways of upregulated genes include extra-cellular matrix [ECM]-receptor interactions and neutrophil extracellular traps, which have been recognized to promote metastasis (*Xiao et al., 2021a*). Downregulated cfRNAs are highly enriched in pathways mainly related to ribosome biogenesis. Downregulation of translation-related pathways was previously reported in tumor-educated platelets (TEPs; *Best et al., 2015*), indicating that translational events might be globally suppressed in the blood milieu of cancer patients. More interestingly, multiple immune-related pathways (PD-1 checkpoint, T-cell differentiation, NOD-like receptor signaling, cytosolic DNA-sensing, and NF-κB signaling) are downregulated in cancer patients, depicting their suppressed immune status. These findings suggest that signals related to the tumor and tumor microenvironment can be identified by cfRNA-seq. For comparisons among different cancer types and HDs, similar patterns were also observed (*Figure 3—figure supplement 1*).

For microbial cfRNAs, we found that the plasma abundance of multiple viral genera, including *Lymphocryptovirus*, *Mastadenovirus*, *Roseolovirus*, several genera of torque teno viruses (TTVs), and *Orthohepadnavirus*, was significantly higher in cancer patients (*Figure 3A*). This result is supported by both pipelines (*Supplementary file 5*). The viral loads of two prevalent genera, *Alphatorquevirus* and *Orthohepadnavirus*, are associated with liver cancer (*Figure 3B*). TTVs are highly prevalent viruses even in the healthy population and are not considered pathogens of a specific disease, but associations between TTV and liver diseases have been widely reported (*Mrzljak and Vilibic-Cavlek, 2020*). Higher TTV abundance is also associated with suppressed immune status and has been utilized as an indicator of immunosuppression after organ transplantation (*Mrzljak and Vilibic-Cavlek, 2020*; *Jaksch et al., 2018*; *Spandole et al., 2015*; *De Vlaminck et al., 2013*). The enrichment of TTVs in cancer patients is concordant with the downregulation of immune pathways we found in human

cfRNAs. The association between liver cancer and *Orthohepadnavirus*, a genus to which HBV belongs, is expected, as 60% of the liver cancer patients in this study had a history of HBV-induced chronic hepatitis (*Supplementary file 1*). Other viral genera that were significantly altered in liver cancer are also shown (*Figure 3C*).

## Evaluating the cancer detection capacity of human and microbial cfRNAs

We used bootstrapping to evaluate the capacity of the plasma cfRNA abundance profile in distinguishing cancer patients from HDs. For both human and microbial cfRNA abundance, we normalized the data and performed batch correction with removing unwanted variations using control genes (RUVg) (*Risso et al., 2014*; *Figure 4—figure supplement 1*). For microbe data, the results of both k-mer-based and alignment-based pipelines were used. Training instances were sampled from the original dataset with replacement until the size of the training set reached the size of the original dataset. Using these training instances, we performed feature selection and fitted a balanced random forest classifier (see Materials and methods). The holdout samples were utilized for performance evaluation. This procedure was repeated 100 times.

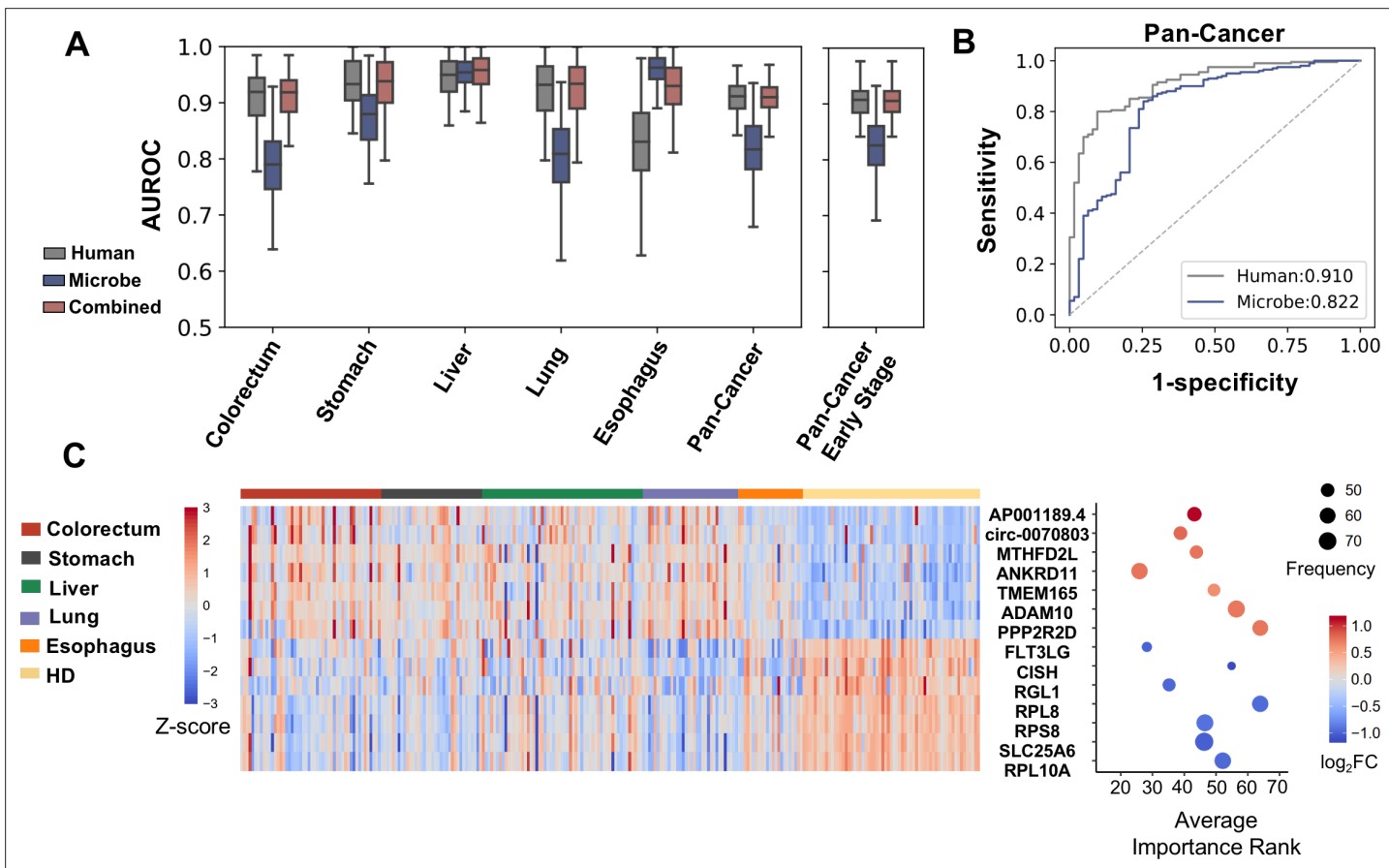

**Figure 4.** Cell-free RNA (cfRNA) features for cancer detection. (**A**) Performance (AUROC) on the holdout dataset in 100 rounds of bootstrap resampling using abundance of human gene expression, microbe abundance (kraken2's results), and combining both data for the binary classification (cancer patients vs. healthy donors). (**B**) Out-of-bag ROC curve using human or microbe features. For each sample, the median value of probabilities predicted by classifiers fitted in bootstrap replicates that reserved this sample in the testing set was utilized to generate the ROC curve. (**C**) Recurrent features with top fold changes when combining human and microbe features for bootstrap analysis. The left panel depicts Z scores of the expression levels in different subjects. The right panel illustrates their average importance ranks, frequency of identified as top 50 features, and fold change compared to healthy donors.

The online version of this article includes the following figure supplement(s) for figure 4:

**Figure supplement 1.** Data normalization for machine learning.

**Figure supplement 2.** Binary classification for cancer detection.

The average AUROC scores of human cfRNAs on testing sets across 100 bootstrap replicates were approximately 0.9, and microbial cfRNAs quantified by k-mer-based pipeline achieved AUROCs from approximately 0.8 to above 0.9 (*Figure 4A*). As the majority of patients in the cohort were in early stages (stages I and II), when only using early-stage cases for bootstrapping, comparable performance was achieved (*Figure 4A*, *Figure 4—figure supplement 2*). A similar result was observed when using the alignment-based method (*Figure 4—figure supplement 2*).

We wondered which features contributed to the model performance in cancer detection. When combining microbe and human features, among those identified as the top 50 most important ones for at least 40 times in 100 bootstrap samplings, features with top fold changes were exemplified (*Figure 4C*). These recurrent features are dominated by human genes. Among the upregulated genes, ADAM10 (encodes a zinc-dependent protease) and TMEM165 (encodes a Golgi body transmembrane protein) have been reported to promote the invasion of tumor cells in multiple cancer types (*Wetzel et al., 2017*; *Smith et al., 2020*; *Lee et al., 2018*). Consistent with our KEGG analysis, the downregulation of several genes that encode protein components of the ribosome (RPL8, RPS8, and RPL10A) in plasma is associated with cancer.

When considering microbial data alone, frequently selected features are shown (*Figure 4—figure supplement 2F*). Compared to human genes, microbial abundance is more heterogeneous in different individuals, which partly explains why microbial features are rarely selected when combined with human gene expression data.

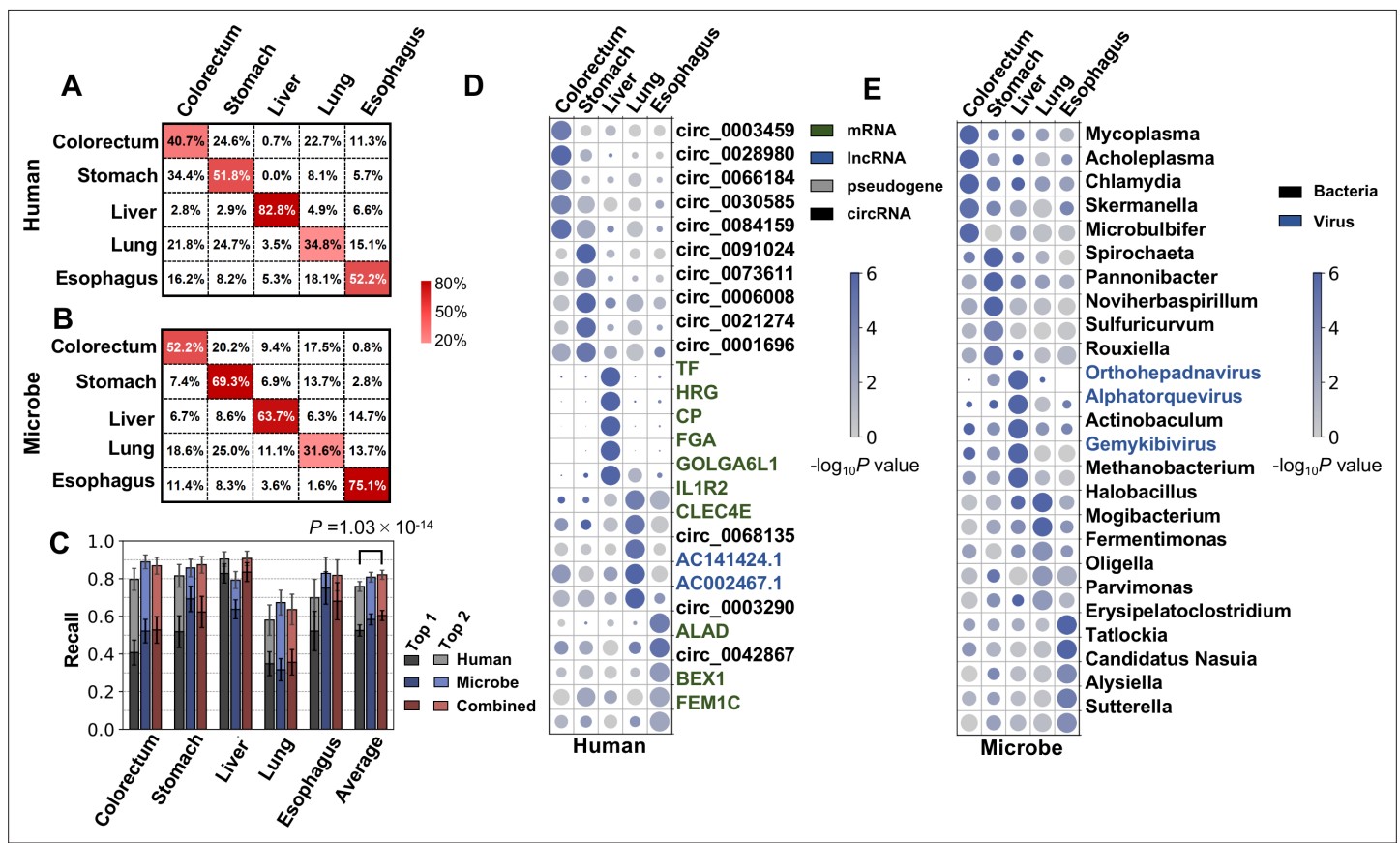

**Figure 5.** Cancer classification using human and microbial cell-free RNAs (cfRNAs). (**A–B**) Confusion matrix of human (**A**) and microbe (**B**) features averaged across bootstrap replicates. (**C**) Top 1 and top 2 recall for each cancer type in multiclass classification. The statistical significance was determined by a one-tailed Mann-Whitney U test. (**D–E**) Recurrent human (**D**) and microbe (**E**) features with the top fold change in multiclass classification. The sizes and colors of the circles indicate the relative abundances (bowtie2 result, scaled to 0–1) and p values in the one vs. rest comparisons, respectively.

The online version of this article includes the following figure supplement(s) for figure 5:

**Figure supplement 1.** Performance for multiclass classification.

## The cancer type specificity of human and microbial cfRNA

Given that the cfRNA profile could distinguish cancer patients from HDs, we further assessed the feasibility of using cfRNAs for classifying cancer patients with different primary tumor locations. A similar bootstrapping strategy was used for performance evaluation.

Using human cfRNA features, an average recall of 52.5% was achieved (*Figure 5A*). The average recall of microbial cfRNA features in the k-mer-based pipeline was 58.4% (*Figure 5B*). These performances were further improved when microbial features were combined with human features: compared to using human features alone, when combining both features, the average recall was 60.4%, improved by 7.9%; the average top 2 recall was 82.1%, improved by 6.2% (*Figure 5C*). Using the alignment-based pipeline, the multiclass classification performance was marginally worse (50.3% on average, *Figure 5—figure supplement 1A*) but still much better than random guesses, and adding microbial features also significantly boosted the average classification performance (*Figure 5—figure supplement 1D*). Taken together, the human and microbial fractions in plasma cfRNAs both provide tumor site-specific information.

Given that cfRNAs can distinguish the primary locations of tumors in cancer patients, some cfRNA features should be specific for certain cancer types. For human and microbe data, we identified features that recurrently ranked as the top 500 most important. Among these recurrent features, for each cancer type, human genes and microbe genera with the greatest fold changes (compared to the remaining cancer types) are illustrated (*Figure 5D*, *Figure 5E*).

For human genes, the top features for colorectal cancer and stomach cancer are mainly circRNAs. Several cfRNAs specific to liver cancer are genes known to be specifically expressed in the liver (TF, HRG, CP, and FGA) (*Liu et al., 2008*). The lung cancer-specific cfRNAs IL1R2 and CLEC4E are related to immune regulation (*Patin et al., 2017*; *Molgora et al., 2018*).

To investigate circRNAs that are specifically upregulated in colorectal cancer and stomach cancer more systematically, we analyzed mioncocirc (*Vo et al., 2019*) data and ranked circRNAs according to fold change between tumor and normal tissue, followed by gene set enrichment analysis (GSEA) using circRNA specifically upregulated. In both cancer types, we found mild but significant enrichment (*Figure 5—figure supplement 1E*), suggesting that a subset of circRNAs upregulated in primary cancer tissue sites may enter the circulatory system and contribute to the plasma cfRNA pool.

Regarding microbial features (*Figure 5E*), *Mycoplasma* and *Acholeplasma* were identified as colorectal cancer specific in our cfRNA profiles. The relevance between *Mycoplasma* infection and cancers was previously reported (*Huang et al., 2001*; *Zella et al., 2018*). *Acholeplasma* was also reported to be more abundant in the gut microbiome of colon cancer patients (*Shoji et al., 2021*). The stomach cancer specific genus *Noviherbaspirillum* was reported to be enriched in oral cancer patients (*Sarkar et al., 2021*). Consistently, *Orthohepadnavirus* and TTVs were again identified as liver cancer specific. *Erysipelatoclostridium*, for which cfRNA is more abundant in the plasma of esophageal cancer patients, is related to several human intestinal diseases (*Sarkar et al., 2021*; *Mancabelli et al., 2017*).

## Discussion

In this study, we sequenced cfRNAs in a cohort of patients with five major types of highly malignant cancer. We demonstrated that there are biologically relevant differences between the cfRNAs of HDs and cancer patients. Cancer type-specific signals could be identified in both human and microbial cfRNAs, and these signals could be utilized to detect and classify multiple cancers, including early-stage cases.

The existence of microbe-derived plasma nucleic acids in donors without sepsis has been independently demonstrated by multiple studies. In typical bioinformatic analysis, reads that cannot be aligned to the human genome are discarded. Our work suggests that these data can be further exploited and provide useful information for microbial profiling in plasma. Several studies have demonstrated that the human virome at different body sites, including plasma, has an unexpected diversity (*Kowarsky et al., 2017*; *Liang and Bushman, 2021*), and current knowledge of human-associated viruses is largely limited to species that could cause severe clinical consequences. Our work highlights the feasibility of discovering clinically relevant but understudied viruses from high-throughput sequencing data.

There are complicated interactions between tumor, the tumor microenvironment, human-associated microbes, and the circulatory system. Tumors with different primary locations have distinct transcriptome compositions and can induce tumor type-specific alterations in other cells or cell fragments, such as TEPs (*Best et al., 2015*). Tumor cells, microbes, and other cells that carry tumor-induced transcriptome alterations all contribute to the cfRNA pool and produce detectable cancer type-specific signals. It is expected that their relative contributions vary in different cancer types. In liver cancer, the identified tumor site-specific features (liver-specific genes and well-known viruses) are readily interpretable. The remaining ones can potentially be explained by the greater contribution of secondary signals that reflect tumor-induced alterations in certain blood components and uncharacterized interactions between humans and microbes. circRNAs have been proposed as exosome-based cancer biomarkers (*Li et al., 2015*). In this study, several plasma circRNAs with cancer type specificity for colorectal and stomach cancer were identified. For colorectal and stomach cancer, the enrichment of upregulated plasma circRNAs suggests that changes in the abundance of plasma circRNAs mirror a subset of circRNA alterations in tumor tissues.

Currently, various cfDNA features (e.g. fragment size, end motif, and methylation) have been well applied to liquid biopsy (*Lo et al., 2021*). Meanwhile, cfRNA provides its own advantages (*Dolgin, 2020*). First, compared to DNAs, many RNAs are more actively transported outside of the cell through carriers such as exosomes; and some cfRNAs, such as the srpRNA RN7SL2, were reported to play regulatory rules in the cancer microenvironment (*Nabet et al., 2017*; *Johnson et al., 2021*). As a result, cfRNA-based biomarkers may provide more functional insights. In addition, RNA expression is tissue-specific; given the dramatic changes in the RNA expression profile in tumors, a fraction of these alterations could be reflected in plasma. Furthermore, the long cfRNA sequencing used in this study detects mRNA of both DNA and RNA viruses, while neither DNA-seq nor small cfRNA-seq can. It has been reported that microbe-derived cfDNA only makes up a small fraction (lower than 0.5% in some cases) of plasma cfDNA (*Zozaya-Valdés et al., 2021*; *Kowarsky et al., 2017*; *Xiao et al., 2021b*). The genomes of bacteria and viruses are much more compact than the human genome, and a larger fraction of their genome sequences are transcribed into RNAs. This indicates that if mixtures of human cells and microbes are sequenced by DNA-seq and RNA-seq to the same depth, microbial reads should make up a larger fraction (approximately 10% on average in our study) in the RNA-seq library, and their signals can be captured more cost-effectively. For these reasons, we believe cfRNA-seq is a cost-effective alternative to cfDNA sequencing, which provides complementary information.

The confounding effect is a major obstacle for discovering reliable biomarkers from high-throughput data. In our cohort design, samples were collected from different clinical centers, and sex for certain cancer types, such as liver cancer, was not well balanced. We attempted to mitigate the problems computationally by using RUVg to remove these unwanted variations. Our analysis provided clues for the clinical relevance of microbe-derived cfRNAs, but a study with a larger, carefully designed cohort is still necessary for clinical application.

## Materials and methods
### Cohort design and sample collection
The cohort in this study included 295 plasma samples in total. Except for 65 previously published samples (GSE142987: 35 liver cancer patients and 30 HDs; *Zhu et al., 2021*), we sequenced the total cfRNAs (>50 nt) in 230 additional plasma samples (54 colorectal cancer, 37 stomach cancer, 27 liver cancer, 35 lung cancer, 31 esophageal cancer, and 46 HDs). The criteria for inclusion were pathologically diagnosed colorectal cancer, stomach cancer, liver cancer, lung cancer, and esophageal cancer patients before surgery, radiation, and chemotherapy.

Samples were obtained between October 2018 and January 2020 from six clinical centers in China: Peking University First Hospital (PKU, Beijing), Peking Union Medical College Hospital (PUMCH, Beijing), Department of Epidemiology Navy Medical University (ShH-1, Shanghai), Eastern Hepatobiliary Surgery Hospital (ShH-2, Shanghai), National Center for Liver Cancer (ShH-3, Shanghai), and Southwest Hospital (SWU, Chongqing). The study was approved by the Peking University First Hospital Biomedical Research Ethics Committee (2018Y15) complied with the declaration of Helsinki. Written informed consent was obtained from all patients prior to the enrollment of this study.

Peripheral whole blood samples were collected from participants before therapy using EDTA-coated vacutainer tubes. The tubes were inverted 8–10 times to mix the blood with anticoagulant. Plasma was separated by a standard clinical blood centrifugation protocol within 2 hr after collection. All plasma samples were aliquoted and stored at –80°C before cfRNA extraction.

## cfRNA-seq library preparation

cfRNAs were extracted from 1 mL of plasma using the Plasma/Serum Circulating RNA and Exosomal Purification kit (Norgen). Purification was based on the use of Norgen's proprietary resin as the separation matrix. This kit extracts all sizes of circulating cfRNAs. The concentration of extracted cfRNAs was assessed using the Qubit RNA assay (Life Technologies).

The total cfRNA library (>50 nt) was prepared with the SMARTer Stranded Total RNA-Seq Kit–Pico. This kit removes ribosomal cDNAs after reverse transcription using a CRISPR/DASH method. We used recombinant DNase I (TAKARA) to digest circulating DNA. ERCC RNA Spike-In Control Mixes (Ambion) were added to the samples before library preparation, with 1 μL per library at an appropriate concentration. RNA Clean and Concentrator-5 kit (Zymo) was used to obtain purified total RNA. More than 20 million reads of total cfRNA were sequenced on an Illumina platform for each library.

Potential contamination in RNA extraction and library preparation was evaluated using two types of negative controls. Two RNA samples were extracted from the *E. coli* DH5α strain, using the same kit for plasma cfRNA extraction. RNA-seq libraries of *E. coli* RNA samples, together with human brain RNA provided by SMARTer Stranded Total RNA-Seq Kit, were constructed using the same protocol for cfRNA library preparation.

## Data processing

For RNA sequencing data, adapters and low-quality sequences in raw sequencing data were trimmed using cutadapt (*Martin, 2011*) (version 2.3). GC oligos introduced in reverse transcription were also trimmed off, and reads shorter than 30 nt were discarded. We used STAR (*Dobin et al., 2013*) (version 2.5.3 a_modified) for sequence mapping. The trimmed reads were sequentially mapped to ERCC's spike-in sequences, vector sequences in NCBI's UniVec database, and human rRNA sequences in RefSeq annotation.

The remaining reads were mapped to the hg38 genome index built with the GENCODE (*Harrow et al., 2012*) v27 annotation. circRNA annotation was downloaded from circBase (*Glažar et al., 2014*). Upstream 150 bp and downstream 150 bp sequences around the back-spliced sites of circRNAs were concatenated to generate junction sequences, and circRNA sequences shorter than 100 bp were discarded. Reads unaligned to hg38 were mapped to circRNA junctions. Duplicates in the aligned reads were removed using Picard Tools MarkDuplicates (version 2.20.0). An aligned read pair was assigned to an RNA type if at least one of the mates overlapped with the corresponding genomic regions. In this way, the aligned reads were sequentially assigned to lncRNAs, mRNAs, snoRNAs, snRNAs, srpRNAs, and Y RNAs with HTSeq (*Anders et al., 2015*) package according to the GENCODE v27 annotation.

The count matrix for human genes was constructed using featureCounts (*Liao et al., 2014*) v1.6.2 with the GENCODE v27 annotation. For downstream analysis, we only considered circRNA junctions annotated in both circBase and mioncocirc (*Vo et al., 2019*). To avoid the impact of potential DNA contamination, only intron-spanning reads were considered.

## Quality control

We filtered cfRNA-seq samples with multiple quality control criteria (*Figure 1—figure supplement 1*): (1) raw reads >10 million; (2) clean reads (reads remained after trimming low quality and adaptor sequences) >5 million; (3) aligned reads after duplicate removal (aligned to the human genome, hg38, and circRNA junctions) >0.5 million; (4) for the clean reads, the fraction of spike-in reads <0.5 and ratio of rRNA reads <0.5; (5) for genome aligned reads, the ratio of mRNA and lncRNA reads >0.2, the ratio of unclassified reads <0.3, and the number of intron-spanning read pairs (defined as a read pair with a CIGAR string in which at least one mate contains 'N' in the BAM files) >100,000.

## Differential analysis and functional enrichment analysis

We used the quasi-likelihood method in the edgeR (*Robinson et al., 2010*) package to identify differentially expressed genes and genera with significant abundance alterations ($|\log_2[\text{fold-change}]|>1$ and FDR <0.05). We used this method to identify differential genes between cancer patients and HDs, as well as genes specific to one cancer type. For cancer type-specific genes, previously reported gender-related genes (*Shi et al., 2019*) were excluded. KEGG pathway enrichment analysis of deregulated genes/RNAs was carried out using clusterProfiler (*Yu et al., 2012*).

## Data normalization

The count matrix of gene expression was normalized using the trimmed mean of M-values (TMM) method in edgeR (*Figure 4—figure supplement 1*). ANOVA was performed among different sample groups (HD and five cancer types) using the quasi-likelihood method in edgeR, and the 25% most insignificant genes that were stably expressed among different groups were considered as empirical control genes. The TMM normalized expression matrix was adjusted by the RUVg function in the RUVSeq (*Risso et al., 2014*) package based on the identified control features.

## Microbial data analysis

Unmapped reads (cleaned reads that failed to align to the human genome or circRNA junctions) were processed independently using a k-mer-based pipeline and an alignment-based pipeline. In the first pipeline, unmapped reads were classified using kraken2 (*Wood et al., 2019*) with its standard database, which contains bacterial, archaeal, viral, and human sequences. In the alignment-based pipeline, using SortMeRNA (*Kopylova et al., 2012*) (version 4.3.3), unmapped reads were annotated as either rRNA or non-rRNA. rRNA reads were mapped to the Silva database with bowtie (*Langmead and Salzberg, 2012*). Non-rRNA reads were aligned to the virus genome curated in kraken2's standard database. In both pipelines, counts at the genus level were used for downstream analysis.

The same preprocessing and downstream analysis pipeline were applied to negative control samples (*E. coli* RNA-seq data were aligned to the reference genome NZ_CP025520.1 with bowtie2, instead of map to human rRNA, human genome, and circRNA junctions). For read coverage analysis of *L. clevelandensis* and HBV, reads unmapped to human sequences were mapped to their reference genomes (NZ_CP012390.1 and NC_003977.2, respectively).

Potential contaminations in genera detected by both the kraken2 pipeline and bowtie2 pipeline (with at least three reads in at least three samples) were filtered prior to downstream analysis. We removed bacterial genera detected in at least one control sample (at least three reads) and virus genera detected in at least one *E. coli* control sample (at least three reads). Genera present in a previously reported common laboratory contamination list (*Salter et al., 2014*) or genera that contain species with counts per million >10 in a published human skin microbiome dataset (*Oh et al., 2016*) were removed. Virus genera that contain species with nonhuman eukaryotic hosts according to virushostdb (*Mihara et al., 2016*) were also excluded. The genera with altered abundance were identified using edgeR. Counts at the genus level were also normalized with TMM and RUVg, as we did for human gene expression.

## Classification performance evaluation

We evaluated the discriminative capacity of cfRNA features with bootstrapping. Training instances were sampled from the full dataset until the sample size of the training set reached the original dataset, and the remaining samples were used for performance evaluation. We used this procedure to generate 100 training sets and corresponding testing sets. For each training set, we performed feature filtering with a rank-sum test. To mitigate the impact of within-class heterogeneity, we sampled a 75% subset of the training instances, performed a rank-sum test (implement withed rank-sums functions in scipy *Virtanen et al., 2020*), nd recorded 50 most significant features, repeated this process 10 times, and took the union of all selected features to fit a balanced random forest classifier (implemented in python package imblearn *Lemaître et al., 2017*). The maximum depths of the trees in the random forest were determined by fivefold cross-validation.

For multiclass classification, a similar bootstrapping strategy was applied. For each of the 100 training-testing pairs, we sampled a 75% subset from the training instances, performed pairwise rank-sum tests, recorded the 50 most significant features, took the union of features selected in

different comparisons, repeated this process 10 times, and took the union of all selected features for model fitting.

## Gene set enrichment analysis

GSEA was implemented with the fgsea (*Korotkevich et al., 2016*) package. For enrichment analysis of circRNA specifically upregulated in one cancer type, circRNA expression data in tumors and normal tissues were downloaded from the mioncocirc (*Vo et al., 2019*) (https://mioncocirc.github.io/) database. For colorectal cancer and esophagus cancer, circRNAs were ranked according to their fold change between tumor and normal tissue, up to 300 circRNAs that were upregulated in one vs. rest comparison with $\log_2$(fold-change) >0.5, and FDR <0.05 were used for enrichment analysis.

## Acknowledgements

This work was supported by the Capital's Funds for Health Improvement and Research (CFH 2022-2-4075), the National Natural Science Foundation of China (31771461, 3217040246, 81972798, 81373067, 81773140, 81902384), the National Key Research and Development Plan of China (2017YFA0505803, 2017YFC0908401, 2019YFC1315700), the National Science and Technology Major Project of China (2018Z × 10723204, 2018Z × 10302205), the Tsinghua University Initiative Scientific Research Program (2021Z99CFY022), the Tsinghua-Foshan Innovation Special Fund, and the Fok Ying-Tong Education Foundation. This study was also supported by the Interdisciplinary Clinical Research Project of Peking University First Hospital, Beijing Advanced Innovation Center for Structural Biology, and the Bioinformatics Platform of National Center for Protein Sciences (Beijing) [2021-NCPSB-005].

## Additional information

### Funding

| Funder | Grant reference number | Author |
| --- | --- | --- |
| the Capital's Fund for Health Improvement and Research | CFH 2022-4075 | Pengyuan Wang |
| National Natural Science Foundation of China | 31771461 | Shanwen Chen |
| the National Key Research and Development Plan of China | 2017YFA0505803 | Zhenjiang Zech Xu |
| the National Science and Technology Major Project of China | 2018ZX10723204 | Pengyuan Wang |
| the Tsinghua University Initiative Scientific Research Program | 2021Z99CFY022 | Zhi John Lu |
| the Tsinghua-Foshan Innovation Special Fund | | Zhi John Lu |
| the Fok Ying-Tong Education Foundation | | Zhi John Lu |
| the Interdisciplinary Clinical Research Project of Peking University First Hospital | | Pengyuan Wang |
| Beijing Advanced Innovation Center for Structural Biology | | Zhi John Lu |

| Funder | Grant reference number | Author |
|---|---|---|
| the Bioinformatics Platform of National Center for Protein Sciences | | Zhi John Lu |
| National Natural Science Foundation of China | 3217040246 | Shanwen Chen |
| National Natural Science Foundation of China | 81972798 | Shanwen Chen |
| National Natural Science Foundation of China | 81373067 | Shanwen Chen |
| National Natural Science Foundation of China | 81773140 | Shanwen Chen |
| National Natural Science Foundation of China | 81902384 | Shanwen Chen |
| the National Key Research and Development Plan of China | 2017YFC0908401 | Zhenjiang Zech Xu |
| the National Key Research and Development Plan of China | 2019YFC1315700 | Zhenjiang Zech Xu |
| the National Science and Technology Major Project of China | 2018ZX10302205 | Pengyuan Wang |

The funders had no role in study design, data collection and interpretation, or the decision to submit the work for publication.

## Author contributions

Shanwen Chen, Conceptualization, Data curation, Investigation, Methodology, Writing - original draft; Yunfan Jin, Formal analysis, Investigation, Methodology, Writing - original draft; Siqi Wang, Data curation, Investigation, Methodology, Writing - original draft; Shaozhen Xing, Investigation, Methodology, Validation, Visualization, Writing - original draft; Yingchao Wu, Shuai Zuo, Yuandeng Luo, Resources; Yuhuan Tao, Yongchen Ma, Investigation, Methodology; Xiaofan Liu, Formal analysis, Investigation, Software; Yichen Hu, Methodology; Hongyan Chen, Project administration, Resources; Feng Xia, Resources, Validation; Chuanming Xie, Project administration, Resources, Validation, Visualization; Jianhua Yin, Resources, Supervision; Xin Wang, Resources, Software, Supervision; Zhihua Liu, Resources, Supervision, Validation; Ning Zhang, Methodology, Resources; Zhenjiang Zech Xu, Conceptualization, Project administration, Writing – review and editing; Zhi John Lu, Conceptualization, Investigation, Methodology, Project administration, Supervision, Writing – review and editing; Pengyuan Wang, Conceptualization, Funding acquisition, Investigation, Methodology, Project administration, Resources

## Author ORCIDs

Zhenjiang Zech Xu http://orcid.org/0000-0003-1080-024X
Pengyuan Wang http://orcid.org/0000-0002-1210-4056

## Ethics

Human subjects: The study was approved by the Peking University First Hospital Biomedical Research Ethics Committee (2018Y15) complied with the declaration of Helsinki. Written informed consent was obtained from all patients prior to the enrollment of this study.

## Decision letter and Author response

Decision letter https://doi.org/10.7554/eLife.75181.sa1
Author response https://doi.org/10.7554/eLife.75181.sa2

## Additional files

### Supplementary files

• Supplementary file 1. Clinical information of samples used in this study.

• Supplementary file 2. Statistics in reads mapping.

• Supplementary file 3. Filtering of potential contamination and read counts in negative control samples.

• Supplementary file 4. Fraction of reads assigned to difference sequences.

• Supplementary file 5. Differential analysis for human and microbial reads between cancer patients and healthy donors.

• Supplementary file 6. Differential analysis for one vs. rest comparisons between different cancer types.

• Supplementary file 7. Recurrency and average importance rank of top 100 features for cancer detection.

• Supplementary file 8. Recurrency and average importance rank of top 500 features for cancer classification.

• Transparent reporting form

### Data availability

Sequencing data have been deposited in GEO under accession codes GSE174302.

The following dataset was generated:

| Author(s) | Year | Dataset title | Dataset URL | Database and Identifier |
|---|---|---|---|---|
| Chen S | 2022 | Cancer type classification using plasma cell-free RNAs derived from human and microbes | http://www.ncbi.nlm.nih.gov/geo/query/acc.cgi?acc=GSE174302 | NCBI Gene Expression Omnibus, GSE174302 |

The following previously published dataset was used:

| Author(s) | Year | Dataset title | Dataset URL | Database and Identifier |
|---|---|---|---|---|
| Zhu Y, Siqi W, Zhi JL | 2020 | RNA-seq analysis of liver cancer patients' plasma | https://www.ncbi.nlm.nih.gov/geo/query/acc.cgi?acc=GSE142987 | NCBI Gene Expression Omnibus, GSE142987 |

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
