## [Editor Report]

This study provides an interesting clinical relevance of human and microbe cell free RNAs derived from plasma that can be used as biomarkers for cancer detection and cancer type classification, and thereby having potential in clinical application.

---

## [Decision Letter]

**Decision letter after peer review:**

Thank you for submitting your article "Cancer Type Classification Using Plasma Cell Free RNAs Derived from Human and Microbes" for consideration by *eLife*. Your article has been reviewed by 2 peer reviewers, and the evaluation has been overseen by YM Dennis Lo as the Senior Editor and Reviewing Editor. The following individual involved in review of your submission has agreed to reveal their identity: Cuncong Zhong (Reviewer #1).

Essential revisions:

1. There have been many genome-wide association studies published in the field for cancer detection and cancer type classification using other technical strategies. What are the main advantages of using cfRNAs as biomarkers to distinguish cancer patients versus healthy donors, comparing with, for example, regular RNA-seq? There are already panels of genes that can be used as biomarkers for cancer patient detection and cancer type classification, based on their expression or other chemical modifications.

2. One of the main challenges in cancer detection is the ability of a technique to identify (or predict) patients at an early stage. The authors profiled samples from a cohort of ~300 patients. What percentage of the patients are at the early stage of the disease, i.e. stage I or II? What is the performance of using the featured cfRNAs as biomarkers to identify cancer patients at early stage? The author should also provide detailed clinical information of the cohort used as a supplementary.

3. In Figure 4, the authors identified cfRNAs in plasma which can be used as "biomarkers" to classify different cancer types, however most of them are circular RNAs. How about the expression of these "biomarkers" in the primary cells from the corresponding tumor types?

4. Microbial contaminations may be introduced during sample preparation. How could one be sure that the microbes detected from this study are actually from patients rather than experimental contaminations?

5. In Figure 5, the authors tried to build a predictive model of cancer detection based on cfRNAs, however, the performance of the predictive model seems very unstable, according to the AUROC, which varies from 0.4-1.0. This result greatly reduces the reliability of the machine learning model. Could they author explain why?

6. The panel of cfRNAs (both human and microbes) itself is more valuable in clinical application than the performance score of the model, if the performance is indeed as good as the authors claimed. Therefore, it will be good if the authors can present this result in more detail.

7. The language of the manuscript can be further improved. The word "some" is overused in many places, which is not scientific.

8. The authors need to redo the classification analysis with bootstrapping.

9. The authors need to correct their errors relating to "validation" and "test" datasets.

10. The authors need to add the patient inclusion criteria.

*Reviewer #1 (Recommendations for the authors):*

Chen et. al studied the use of cfRNA data for the diagnosis of five types of cancer. The authors claim that cfRNA reads from both human and microbial sources can both contribute to the cancer diagnosis. Overall, I find the hypothesis and design of the experiment reasonable. If successful, the results of this work can be used to detect major cancer types in early stages, significantly improving the survival rate of the cancer patients.

The authors need to redo the classification analysis with boosting.

The authors need to correct their errors relating to "validation" and "test" datasets.

The authors need to add the patient inclusion criteria.

*Reviewer #2 (Recommendations for the authors):*

In this manuscript, Chen et al. presents clinical relevance of human and microbe cfRNAs derived from plasma, which can be used as biomarkers for cancer detection and cancer type classification. The authors profiled cfRNAs in a cohort of ~300 plasma samples of five cancer types (colorectal cancer, stomach cancer, liver cancer, lung cancer, esophageal cancer) and healthy donors with RNA-seq. They claimed a prediction rate over 0.9 for distinguishing cancer patients from healthy donors and an improvements of predictive power in cancer type classification by combining human and microbial features together. Overall, I found this study interesting and potentials in clinical application. The experiment is well designed and data properly analyzed. The presented results also support the conclusions. The motivation and potential impacts of the this study would need to be further addressed the Introduction and Discussion sections.

My main concerns are as follows:

1. There have been many genome-wide association studies published in the field for cancer detection and cancer type classification using other technical strategies. What are the main advantages of using cfRNAs as biomarkers to distinguish cancer patients versus healthy donors, comparing with, for example, regular RNA-seq? I believe there are panels of regular genes that can be used as biomarkers for cancer patient detection and cancer type classification, based on their expression or other chemical modifications.

2. One of the main challenges in cancer detection is the ability of a technique to identify (or predict) patients at an early stage. The authors profiled samples from a cohort of ~300 patients, and I am wondering what percentage of the patients are at the early stage of the disease, i.e. stage I or II? What is the performance of using the featured cfRNAs as biomarkers to identify cancer patients at early stage? The author should also provide detailed clinical information of the cohort used as a supplementary.

3. In Figure 4, the authors identified cfRNAs in plasma which can be used as "biomarkers" to classify different cancer types, however most of them are circular RNAs. How about the expression of these "biomarkers" in the primary cells from the corresponding tumor types?

4. Microbial contaminations may be introduced during sample preparation. I am not convinced whether the microbes detected from this study are actually from patients rather than experimental contaminations.

5. In Figure 5, the authors tried to build a predictive model of cancer detection based on cfRNAs, however, the performance of the predictive model seems very unstable, according to the AUROC, which varies from 0.4-1.0. This result greatly reduces the reliability of the machine learning model. Could they author explain why?

6. The panel of cfRNAs (both human and microbes) itself is more valuable in clinical application than the performance score of the model, if the performance is indeed as good as the authors claimed. Therefore, it will be good if the authors can present this result in more detail.

7. The language of the manuscript can be further improved. The word "some" is overused in many places, which is not scientific.

---

## [Author Response]

Essential revisions:1. There have been many genome-wide association studies published in the field for cancer detection and cancer type classification using other technical strategies. What are the main advantages of using cfRNAs as biomarkers to distinguish cancer patients versus healthy donors, comparing with, for example, regular RNA-seq? There are already panels of genes that can be used as biomarkers for cancer patient detection and cancer type classification, based on their expression or other chemical modifications.

We thank the editor and reviewer for this comment. We add the following text in Introduction:

“Most of the previous cfRNA studies focused on small RNA species^20^, which are relatively stable in plasma. Long RNA species in plasma have relatively low concentrations, which are mainly 100-200 nt fragments lacking poly-A tails and intact ends. Therefore, regular RNA-seq, which usually uses ligation techniques to add adaptors, will not work well for long cfRNAs. The recently developed SMART-seq^21^ based techniques offer the potential to overcome these issues. Furthermore, to sequence total RNAs in plasma, we need to simultaneously remove the abundant rRNA fragments, which are enabled by a CRISPR-based technology called Depletion of Abundant Sequences by Hybridization (DASH)^22^. This motivated us to study the biological relevance and clinical utilities of human and microbe-derived long cfRNAs, taking advantage of the above techniques.”

We also add the following text in Discussion:

“Currently, various cfDNA features (e.g., fragment size, end motif, methylation) have been well applied to liquid biopsy^58^. Meanwhile, cfRNA provides its own advantages^59^. First, compared to DNAs, many RNAs are more actively transported outside of the cell through carriers such as exosomes; and some cfRNAs, such as the srpRNA RN7SL2, were reported to play regulatory rules in the cancer microenvironment^60,61^. As a result, cfRNA-based biomarkers may provide more functional insights. In addition, RNA expression is tissue-specific; given the dramatic changes in the RNA expression profile in tumors, a fraction of these alterations could be reflected in plasma. Furthermore, the long cfRNA sequencing used in this study detects mRNA of both DNA and RNA viruses, while neither DNA-seq nor small cfRNA-seq can. It has been reported that microbe-derived cfDNA only makes up a small fraction (lower than 0.5% in some cases) of plasma cfDNA^15,16,62^. The genomes of bacteria and viruses are much more compact than the human genome, and a larger fraction of their genome sequences are transcribed into RNAs. This indicates that if mixtures of human cells and microbes are sequenced by DNA-seq and RNA-seq to the same depth, microbial reads should make up a larger fraction (approximately 10% on average in our study) in the RNA-seq library, and their signals can be captured more cost-effectively. For these reasons, we believe cfRNA-seq is a cost-effective alternative to cfDNA sequencing, which provides complementary information.”

2. One of the main challenges in cancer detection is the ability of a technique to identify (or predict) patients at an early stage. The authors profiled samples from a cohort of ~300 patients. What percentage of the patients are at the early stage of the disease, i.e. stage I or II? What is the performance of using the featured cfRNAs as biomarkers to identify cancer patients at early stage? The author should also provide detailed clinical information of the cohort used as a supplementary.

We thank the reviewer for this suggestion. We have provided detailed clinical information of the cohort in Supplementary Table 1. In cohort design, we’ve focused on early cancer detection: for five cancer types, the majority of patients were in early stages (stage I and stage II): 83% (45 in 54) for colorectal cancer, 65% (24 in 37) for stomach cancer, 71% (44 in 62) for liver cancer, 86% (30 in 35) for lung cancer, and 71% (22 in 31) for esophageal cancer. We also evaluated the performance of our model on early-stage patients (Figure 4 A, Figure 4—figure supplement 2A-E). As expected, since the majority of patients were in the early stage, the model performance on early-stage patients was comparable with that on the whole cohort.

3. In Figure 4, the authors identified cfRNAs in plasma which can be used as "biomarkers" to classify different cancer types, however most of them are circular RNAs. How about the expression of these "biomarkers" in the primary cells from the corresponding tumor types?

We thank the reviewer for this comment. We have looked into the expression of these biomarkers in cells/tissues. (Please note that we have merged the previous Figure 4 to the new Figure 5 in order to better address comment 6 of the editor.)

As mentioned by the reviewer, most cancer type-specific cfRNAs identified in colorectal cancer and stomach cancer were circRNAs, we further investigate the expression of circRNAs that are specifically up-regulated in these two cancer types in tumor tissue using mioncocirc (Vo et al., 2019) data, and observed significant enrichment in both cancer types. We add the following statements in our manuscripts:

“To investigate circRNAs that are specifically upregulated in colorectal cancer and stomach cancer more systematically, we analyzed mioncocirc^52^ data and ranked circRNAs according to fold change between tumor and normal tissue, followed by gene set enrichment analysis (GSEA) using circular RNA specifically up-regulated. In both cancer types, we found mild but significant enrichment (Figure 5—figure supplement 1E), suggesting that a subset of circRNAs upregulated in primary cancer tissue sites may enter the circulatory system and contribute to the plasma cfRNA pool.”

4. Microbial contaminations may be introduced during sample preparation. How could one be sure that the microbes detected from this study are actually from patients rather than experimental contaminations?

We thank the reviewer for this important comment. We did carefully prevent the bias of contamination in several ways. As shown in Figure 1 A, we combined experimental and computational methods to mitigate the impact of potential contaminations. We sequenced three samples (2 *E. coli* RNA samples and 1 human brain RNA sample) as negative controls for RNA extraction and library preparation, to determine identifiable lab contaminants. We also curated a list of published common lab contaminants and a list of abundant skin microbes (which could be introduced in sample collection). Microbes detected in control samples and curated potential contaminations were combined into a black list, and genera in this list were excluded from downstream analysis. We believed that these filters allow us to exclude most of the potential contaminations. For instance, some viruses with differential abundance, like HBV and TTV, are indeed biologically relevant.

5. In Figure 5, the authors tried to build a predictive model of cancer detection based on cfRNAs, however, the performance of the predictive model seems very unstable, according to the AUROC, which varies from 0.4-1.0. This result greatly reduces the reliability of the machine learning model. Could they author explain why?

This variability can be largely explained by the small sample size for performance evaluation in cross-validation. Therefore, following the suggestion of another reviewer (see Comments 8 of the editor), we redid the classification with bootstrapping. In each bootstrap replicate, we sampled training instances from the whole dataset with replacement, until the number of training instances reached the size of the original dataset. Now the testing set (instances that are not sampled in bootstrap) gets larger than the that in our previous version. For colorectal cancer vs. HD comparison, 50 colorectal cancer and 63 HD samples passed quality control, in theory, the size of the testing set is (1-0.632)*(50+63)≈42, much larger than the previous testing set (around 12 samples), and variability in classification performance greatly reduced (Figure 4 A, Figure 4—figure supplement 2).

6. The panel of cfRNAs (both human and microbes) itself is more valuable in clinical application than the performance score of the model, if the performance is indeed as good as the authors claimed. Therefore, it will be good if the authors can present this result in more detail.

We thank the reviewer for this suggestion. To better illustrate features that contribute to classification performance, we reorganized our figures. We split the original figure 5 (classification performance) into two figures, one for binary classification (now figure 4), one for multiclass classification (now figure 5), and merged the original figure 4 (cancer type-specific features) to figure 5. In general, the contribution of microbial features was more important in multiclass classification (new figure 5) than that in binary classification (new figure 4).

For pan-cancer vs. HD binary classification, we illustrated features that were recurrently identified as the top-ranked when combining human gene expression and microbe abundance in bootstrap analysis (new Figure 4C). We added the following statements in our manuscript:

“These recurrent features are dominated by human genes. Among the upregulated genes, ADAM10 (encodes a zinc-dependent protease) and TMEM165 (encodes a Golgi body transmembrane protein) have been reported to promote the invasion of tumor cells in multiple cancer types^46-48^. Consistent with our KEGG analysis, the downregulation of several genes that encode protein components of the ribosome (RPL8, RPS8, and RPL10A) in plasma is associated with cancer.”

As these features were dominated by human genes, when only considering microbe features, we illustrated recurrent features for pan-cancer binary classification in Figure 4—figure supplement 2F. Frequency and importance of identified features in other binary classifications (one cancer type vs. HD) were added to supplementary material.

In between cancer type classifications, we highlight the performance improvement with different color scale when adding microbial features (new Figure 5A-B). For features recurrently identified as top features in bootstrapping, we exemplified the ones with largest fold changes (new Figure 5D-E). Previously we only required the features to have large fold changes (old Figure 4), without the prediction recurrence constraint. We believe that the new figure better illustrated the features contributing to the prediction model.

7. The language of the manuscript can be further improved. The word "some" is overused in many places, which is not scientific.

We thank the reviewer for this suggestion. We’ve replaced the overused words. We’ve also hired a professional service to edit the English.

8. The authors need to redo the classification analysis with bootstrapping.

We thank the reviewer for this constructive suggestion. We fully agree that we cannot determine the variability in model performance with a single randomly sampled testing set, especially given the relatively small sample size. We have rerun all of the classifications with bootstrapping. In each bootstrap replicate, training instances were sampled from the original dataset with replacement, and were used for feature selection and model training. The holdout samples were used for performance evaluation. The updated results were shown in Figure 4, Figure 5, Figure 4—figure supplement 2 and Figure 5—figure supplement 1. Our previous conclusion still holds. Meanwhile, this bootstrapping procedure greatly reduces the variability of our model performance.

9. The authors need to correct their errors relating to "validation" and "test" datasets.

We thank the reviewer for pointing out this, we have changed our notation for different datasets. Now we clarified the nomenclature in the revised manuscript and replaced “validation set” with “test set”.

10. The authors need to add the patient inclusion criteria.

We thank the reviewer for this suggestion. We’ve added the following patient inclusion criteria in method section:

“The criteria for inclusion were pathologically diagnosed colorectal cancer, stomach cancer, liver cancer, lung cancer and esophageal cancer patients before surgery, radiation and chemotherapy.”

Reviewer #1 (Recommendations for the authors):Chen et. al studied the use of cfRNA data for the diagnosis of five types of cancer. The authors claim that cfRNA reads from both human and microbial sources can both contribute to the cancer diagnosis. Overall, I find the hypothesis and design of the experiment reasonable. If successful, the results of this work can be used to detect major cancer types in early stages, significantly improving the survival rate of the cancer patients.

We thank the reviewer for the overall positive comments and constructive suggestions. The editor has summarized the major points as comments 8-10 in “essential revisions”, please refer to comments 8-10 in “Response to editor” section for point-to-point responses.

The authors need to redo the classification analysis with boosting.

Please see response to comment 8 of the editor.

The authors need to correct their errors relating to "validation" and "test" datasets.

Please see response to comment 9 of the editor.

The authors need to add the patient inclusion criteria.

Please see response to comment 10 of the editor.

Reviewer #2 (Recommendations for the authors):In this manuscript, Chen et al. presents clinical relevance of human and microbe cfRNAs derived from plasma, which can be used as biomarkers for cancer detection and cancer type classification. The authors profiled cfRNAs in a cohort of ~300 plasma samples of five cancer types (colorectal cancer, stomach cancer, liver cancer, lung cancer, esophageal cancer) and healthy donors with RNA-seq. They claimed a prediction rate over 0.9 for distinguishing cancer patients from healthy donors and an improvements of predictive power in cancer type classification by combining human and microbial features together. Overall, I found this study interesting and potentials in clinical application. The experiment is well designed and data properly analyzed. The presented results also support the conclusions. The motivation and potential impacts of the this study would need to be further addressed the Introduction and Discussion sections.

We thank the reviewer for the positive feedback and constructive suggestions. We’ve added more discussion on motivation and potential impacts of this study in the introduction and Discussion section.

In introduction section:

“As total RNA-seq captures RNA fragments regardless of their origination, profiling total cfRNAs should provide rich information for both human and microbe transcripts. …”.

In Discussion section:

“We demonstrated that there are biologically relevant differences between cfRNA of healthy donors and cancer patients…Our work highlights the feasibility of discovering clinically relevant but understudied viruses from high throughput sequencing data…we believe cfRNA-seq is a cost-effective alternative to cfDNA sequencing, which provides complementary information.”

The remaining 7 major points were summarized as comments 1-7 by the editor in the essential revisions list, please refer to comments 1-7 in “Response to editor” section for point-to-point responses.

My main concerns are as follows:1. There have been many genome-wide association studies published in the field for cancer detection and cancer type classification using other technical strategies. What are the main advantages of using cfRNAs as biomarkers to distinguish cancer patients versus healthy donors, comparing with, for example, regular RNA-seq? I believe there are panels of regular genes that can be used as biomarkers for cancer patient detection and cancer type classification, based on their expression or other chemical modifications.

Please see response to comment 1 of the editor.

2. One of the main challenges in cancer detection is the ability of a technique to identify (or predict) patients at an early stage. The authors profiled samples from a cohort of ~300 patients, and I am wondering what percentage of the patients are at the early stage of the disease, i.e. stage I or II? What is the performance of using the featured cfRNAs as biomarkers to identify cancer patients at early stage? The author should also provide detailed clinical information of the cohort used as a supplementary.

Please see response to comment 2 of the editor.

3. In Figure 4, the authors identified cfRNAs in plasma which can be used as "biomarkers" to classify different cancer types, however most of them are circular RNAs. How about the expression of these "biomarkers" in the primary cells from the corresponding tumor types?

Please see response to comment 3 of the editor.

4. Microbial contaminations may be introduced during sample preparation. I am not convinced whether the microbes detected from this study are actually from patients rather than experimental contaminations.

Please see response to comment 4 of the editor.

5. In Figure 5, the authors tried to build a predictive model of cancer detection based on cfRNAs, however, the performance of the predictive model seems very unstable, according to the AUROC, which varies from 0.4-1.0. This result greatly reduces the reliability of the machine learning model. Could they author explain why?

Please see response to comment 5 of the editor.

6. The panel of cfRNAs (both human and microbes) itself is more valuable in clinical application than the performance score of the model, if the performance is indeed as good as the authors claimed. Therefore, it will be good if the authors can present this result in more detail.

Please see response to comment 6 of the editor.

7. The language of the manuscript can be further improved. The word "some" is overused in many places, which is not scientific.

Please see response to comment 7 of the editor.